# Uveal Melanoma Patients Have a Distinct Metabolic Phenotype in Peripheral Blood

**DOI:** 10.3390/ijms24065077

**Published:** 2023-03-07

**Authors:** Daniël P. de Bruyn, Michiel Bongaerts, Ramon Bonte, Jolanda Vaarwater, Magda A. Meester-Smoor, Robert M. Verdijk, Dion Paridaens, Nicole C. Naus, Annelies de Klein, George J. G. Ruijter, Emine Kiliç, Erwin Brosens

**Affiliations:** 1Department of Ophthalmology, Erasmus MC, 3000 CA Rotterdam, The Netherlands; 2Department of Clinical Genetics, Erasmus MC, 3000 CA Rotterdam, The Netherlands; 3Erasmus MC Cancer Institute, Erasmus MC, 3000 CA Rotterdam, The Netherlands; 4The Rotterdam Eye Hospital, 3011 BH Rotterdam, The Netherlands; 5Department of Pathology, Section Ophthalmic Pathology, Erasmus MC, 3000 CA Rotterdam, The Netherlands; 6Department of Pathology, Leiden University Medical Center, 2333 ZA Leiden, The Netherlands

**Keywords:** noninvasive, prognostication, biomarker, metabolomics, liquid biopsy

## Abstract

Uveal melanomas (UM) are detected earlier. Consequently, tumors are smaller, allowing for novel eye-preserving treatments. This reduces tumor tissue available for genomic profiling. Additionally, these small tumors can be hard to differentiate from nevi, creating the need for minimally invasive detection and prognostication. Metabolites show promise as minimally invasive detection by resembling the biological phenotype. In this pilot study, we determined metabolite patterns in the peripheral blood of UM patients (n = 113) and controls (n = 46) using untargeted metabolomics. Using a random forest classifier (RFC) and leave-one-out cross-validation, we confirmed discriminatory metabolite patterns in UM patients compared to controls with an area under the curve of the receiver operating characteristic of 0.99 in both positive and negative ion modes. The RFC and leave-one-out cross-validation did not reveal discriminatory metabolite patterns in high-risk versus low-risk of metastasizing in UM patients. Ten-time repeated analyses of the RFC and LOOCV using 50% randomly distributed samples showed similar results for UM patients versus controls and prognostic groups. Pathway analysis using annotated metabolites indicated dysregulation of several processes associated with malignancies. Consequently, minimally invasive metabolomics could potentially allow for screening as it distinguishes metabolite patterns that are putatively associated with oncogenic processes in the peripheral blood plasma of UM patients from controls at the time of diagnosis.

## 1. Introduction

Uveal melanoma (UM) affects 2–9 per million persons yearly. In UM, three mutually exclusive secondary driver mutations occur in BRCA-associated protein 1 (*BAP1*), splicing factor 3 subunit B1 (*SF3B1*), and eukaryotic translation initiation factor 1A, X-linked (*EIF1AX*), and their corresponding copy number variation profiles are highly correlated to prognosis [1]. Loss-of-function mutations in the tumor suppressor gene *BAP1* are correlated with large structural chromosomal aberrations, including the loss of chromosome 3. Other chromosomal aberrations include 1p loss and isochromosome 6p or 8q formation, with a corresponding loss of chromosome 6q and gain of chromosome 8q [1]. These “BAP1 negative” tumors are associated with a metastatis-free survival of less than five years [2]. *SF3B1* is involved in the spliceosome complex, and *SF3B1*-mutated UMs frequently have smaller telomeric chromosomal aberrations and a partial gain of chromosome 8q. These tumors pose an intermediate risk of metastasizing, and metastases arise within 5 to 15 years [3]. EIF1AX-mutated tumors usually occur with a concurrent gain of chromosome 6p, and these tumors hardly ever metastasize [3]. The *SF3B1* and *EIF1AX* secondary driver mutations occur with disomy 3 in UM [2,3].

Improved detection methods result in earlier diagnoses and an increase in the detection of small lesions [4,5]. Treatment of these smaller primary lesions has shifted from surgical intervention to eye-preserving therapies [4,6]. While preserving the patient’s eye, this strategy generally reduces the amount of available tumor material for molecular prognostication. Additionally, diagnosing UM requires specialized ophthalmologic expertise. Patients harboring a UM often experience a referral delay to an academic ophthalmological center to confirm the UM diagnosis [7]. Both of these trends create the need for easy-to-use detection and prognostication modalities.

Blood-based biomarkers have the potential to replace invasive characterization, but the abundance of tumor biomarkers in peripheral blood, such as circulating tumor cells [8], circulating tumor DNA, and exosomes [9], is often low. Dysregulation of specific metabolic pathways is associated with malignant processes [10], and the products of these altered pathways are detected in the blood of patients harboring varying malignancies, such as lung adenocarcinoma [11] and colorectal cancer [12]. Untargeted metabolomics allows for hypothesis-free biomarker detection, and plasma metabolome studies show promising results in the early detection of different types of cancer [11,13]. In this study, we evaluate the potential of untargeted metabolomics to discriminate UM patients from control participants and evaluate if metabolite patterns can distinguish the prognostically relevant UM patient subgroups.

## 2. Results

We determined metabolite abundances obtained from the peripheral blood of 159 participants in a discovery (n = 114) and replication (n = 45) cohort, including 110 samples of UM patients diagnosed between 1998 and 2021 (MEC-2009-375) and 46 samples from controls obtained between 2016 and 2017 from the Combined Ophthalmic Research Rotterdam Biobank (CORRBI) (MEC-2012-031) (Table 1 and Appendix A). The discovery cohort consisted of samples from 37 *BAP1*, 16 *SF3B1*, 15 *EIF1AX* patients, and 46 non-UM controls. The replication cohort consisted of samples from 19 *BAP1*, 17 *SF3B1*, and 9 *EIF1AX* patients. The average storage time of blood samples from controls and UM patients was similar (*p* > 0.05). However, blood samples of controls were stored for shorter than five years, whereas storage times of samples from UM patients ranged between 0 and 20 years. UM patients and controls were comparable regarding sex distribution, comorbidities, and medication use. The control cohort comprised patients with ophthalmic diseases common in the elderly, and individuals were on average slightly older compared to the patients in the UM patient cohort (73.9 years vs. 63.0 years, *p* < 0.0001). UM-subgroups were homogeneous regarding tumor size, location, and primary driver mutation. Other prognostic factors were more prevalent in *BAP1*-mutated tumors (Table 1), which conform to the characteristics of these three UM subclasses [1].

In order to determine metabolite abundancies in peripheral blood, we performed ultra high-pressure liquid chromatography-orbitrap mass spectrometry (UHPLC-MS) in a discovery and replication cohort (Appendix A). Multiple quality control methods were assessed. Metabolite abundancies were normalized using internal standards (Appendix A). Metchalizer [14] was used for the normalization of the merged dataset, and no batch effects were observed using Metchalizer (Appendix A). Correlation analyses using biological and technical replicates (Appendix A) indicated good concordance in the discovery cohort and merged dataset, respectively (Appendix A). In untargeted metabolomics, a high number of metabolite abundancies is tested, resulting in a high number of *p*-values. The ideal distribution of *p*-values has a peak of *p*-values in the left tail (meaning lower values for *p*) and a uniform distribution for null *p*-values. We plotted the distribution of *p*-values obtained from the *t*-test for metabolite abundancies between UM and controls and observed the ideal distribution with a subset of differentially abundant metabolites, and *p*-values for not differentially abundant metabolites are uniformly distributed in the positive and negative ion modes (Appendix A). Unsupervised two and three-dimensionality reduction analysis methods did not show systemic differences between the different groups in the positive and negative ion modes (Appendix A). We also evaluated the influence of the different common diseases in the elderly that were present in our control cohort on the differences found in metabolite patterns. The control cohort clustered together, and no different sub-clusters could be established in the three dimensions of the principal component analysis (PCA) or in the two dimensions of the t-distributed stochastic neighbor embedding (t-SNE) analysis within the control cohort in the positive and negative ion modes (Appendix A). The control cohort mainly comprised people harboring age-related diseases that frequently arise in the population. The prevalence of age-related macular degeneration (AMD) in Europe ranges from 3.5% to 9.8% for people older than 55 years [15]. Cataract is one of the most common causes of visual impairment that comes with age [16]. Prior to visual impairment, the lens becomes cloudy, and the prevalence of impaired cataracts in people over 60 worldwide is 54.4% [17]. UM occurs in the elderly, and our UM patient cohort resembles that. Therefore, we could expect UM patients to harbor (not visually impairing) cataract and AMD, just as the general population does. Medication use in the elderly is common, and in our control cohort, we observed similar medication use as in our UM patient cohort. Therefore, we believe our control cohort is representative of non-UM controls over 60 years of age.

### 2.1. Untargeted Metabolomics Discriminates UM Patients from Control Participants

UM patients have distinctly different metabolite patterns compared to controls. Metabolite profiles are analyzed in both ion modes to increase the confidence in our findings, as metabolites can have an ionization preference for either protonation or deprotonation. We evaluated the discriminatory power of metabolite patterns using a partial least squares-determinant analysis (PLS-DA), in which a clear distinction is observed between metabolite patterns derived from peripheral blood of UM patients and controls in the discovery cohort and merged dataset (Figure 1A,C). Next, a random forest classifier (RFC) was trained and validated using a leave-one-out cross-validation (LOOCV) procedure, classifying UM patients versus controls. LOOCV in the discovery cohort resulted in a precision of 0.96 and 1.00 and a recall of 0.79 and 0.94 in the positive and negative ion modes, respectively (Table 2). We visualized the sensitivity and specificity of the trained models, as observed using the LOOCV procedure, by the area under the curve (AUC) of the receiver operating characteristic (ROC) in positive and negative ion modes for the discovery cohort as well as the merged dataset. The AUCs of the ROCs in the discovery cohort were 0.99 and 0.97 for the positive and negative ion modes, respectively (Figure 1B). The AUCs of the ROCs in the merged dataset are 0.99 and 0.99 in the positive and negative ion modes, respectively (Figure 1D). To evaluate the robustness of the discriminative power for distinguishing UM patients from controls, we repeated this LOOCV procedure 10 times by training on a dataset where samples were randomly selected and evenly distributed per group in two batches (Appendix A). These in silico replicated experiments showed good performance of the RFCs in both the positive and negative ion modes. For each of these LOOCVs, we obtained AUCs of ROCs that range from 0.953 to 0.998 (Appendix A).

### 2.2. Metabolic Patterns Do Not Distinguish Prognostic Subtypes at the Time of Diagnosis

UMs can metastasize prior to diagnosis and, consequently, can present after the primary tumor is removed or irradiated [1,18]. These micro-metastases can come out of dormancy, and most patients harboring a *BAP1* mutation will develop metastases within 60 months. Patients harboring an *SF3B1* mutation will develop metastases within 120 months [1]. Additionally, in the primary driver gene *GNA11*, mutations are associated with a shorter metastatis-free survival and a loss of BAP1 expression [19]. We visualized differences in metabolite patterns in the molecular subclasses (*BAP1*, *SF3B1,* and *EIF1AX*) and, more specifically, the prognostically poor and favorable secondary driver mutations, *BAP1* and *EIF1AX*, using supervised dimensionality reduction analyses. A moderate separation based on metabolite patterns is seen between the three molecular subclasses (Figure 2A), which is more profound in the PLS-DA comparing only the metabolite patterns of patients harboring a *BAP1* or *EIF1AX*-mutated tumor (Figure 2B). A PLS-DA easily provides differences between the analyzed groups; therefore, we deployed more robust techniques to evaluate the discriminatory power of metabolite abundancies. We have used an RFC with LOOCV procedure as it provides a reliable impression of the found discriminatory metabolite patterns. Unfortunately, we could not confirm the differences found in the PLS-DA using an RFC and LOOCV procedure in the merged dataset, nor could we confirm differential metabolite patterns between the molecular subclasses (Table 3).

F1-scores resemble the harmonic mean of precision and recall and are thus the metric for the combined performance of the classifier. The trained RFC performs poorly in the discovery and replication cohort, meaning that a distinction between secondary driver mutations cannot be made based on metabolite patterns (Table 3A,B). The performance of the RFC is assessed by the LOOCV procedure, and the F1-scores were 0.60, 0.00, and 0.00 for patients harboring a *BAP1*, *SF3B1,* and *EIF1AX*-mutated tumor in the negative ion mode, respectively; and 0.53, 0.00, and 0.00 for patients harboring a *BAP1*, *SF3B1,* and *EIF1AX*-mutated tumor in the positive ion mode, respectively (Table 3A). In the replication cohort, metabolite abundancies are slightly different in the *BAP1* group, with F1-scores of 0.82, 0.59, and 0.59 for patients harboring a *BAP1*, *SF3B1,* and *EIF1AX*-mutated tumor in the negative ion mode, respectively (Table 3B). In the positive ion mode, F1-scores were 0.65, 0.35, and 0.64 for patients harboring a *BAP1*, *SF3B1,* and *EIF1AX*-mutated tumor, respectively (Table 3B). In the meta-analysis, the RFC on the merged dataset also performed poorly (Table 3C). However, controls are detected, even when the three mutational groups are selected and show differential metabolite patterns (Table 3C).

As the primary driver mutations, *GNAQ* and *GNA11*, also provide prognostic relevance [19], we deployed an RFC which performed poorly (F1-score: 0.57 and 0.61 for *GNAQ* and *GNA11* prediction, respectively, in the negative ion mode. F1-score: 0.57 and 0.61 for *GNAQ* and *GNA11* prediction, respectively, in the positive ion mode) (Table 3D). Furthermore, converse to the supervised discriminant analyses, the LOOCV approach does not show differences in metabolite patterns between patients harboring a *BAP1* or *EIF1AX*-mutated tumor in the merged dataset (Table 3E). Ultimately, we trained an RFC to classify patients based on the future development of metastases but could not establish a sufficient prediction model as the LOOCV shows that differences in metabolite patterns are insufficient to distinguish patients by metastatic formation. The LOOCV procedure showed F1-scores of 0.23 and 0.82 for the detection of metastatic and localized disease in the negative ion mode, respectively, and F1-scores of 0.15 and 0.80 for the detection of metastatic and localized disease in the positive ion mode, respectively (Table 3F). In order to confirm the discriminatory power of differential metabolite patterns, we in silico repeated these analyses ten times. This ten-time bootstrap entails using all samples of the merged dataset and randomly picking 50% of the samples in the *BAP1*, *SF3B1*, *EIF1AX,* and control groups and using these for training an RFC and evaluating the model with the LOOCV procedure (Appendix A). This ten-time bootstrapped method showed poor discriminatory power of metabolite patterns to distinguish prognostically relevant subgroups, similar to the original models (Appendix A).

### 2.3. Differentially Expressed Pathways in Peripheral Blood of Uveal Melanoma Patients

To further investigate the metabolic differences between UM patients and controls, we correlated feature abundancies to tumor size. Moreover, we detected features that were differentially expressed in UM patients when compared to the control cohort using the t-statistic. The t-statistic and correlation coefficient (when correlated to tumor size) were used to explore regions where both metrics were significant since that would indicate features that were associated with UM (Appendix A). However, none were significant after multiple testing corrections (Appendix A).

Next, we investigated differentially expressed pathways using in-house annotated metabolites, as described previously [20]. We observed an upregulation of transfer RNA (tRNA) charging and glycine usage for creatine biosynthesis (Appendix A and Appendix A). Purine ribonucleosides degradation and the super pathway of citrulline metabolism were downregulated in UM patients (Appendix A and Appendix A). tRNA charging generally promotes oncogenesis [21] and, consistent with the upregulation of tRNA charging, we observed a lower abundance of the amino acids arginine, asparagine, cysteine, leucine, lysine, methionine, phenylalanine, proline, tryptophan, tyrosine, and valine in UM patients. Glycine is more abundant in the peripheral blood of UM patients and extracellular glycine is internalized by cells and degraded to produce substrates for nucleotides, proteins, glutathione, and methylation [22]. Additionally, we observed a lower abundance of adenine, guanine, and D-ribose-5-phosphate (Appendix A), which indicates either a downregulation of purine degradation or increased salvage of bases and can indicate a higher energy consumption. Guanine is metabolized for recycling bases by hypoxanthine-guanine phosphoribosyltransferase, which metabolizes guanine into guanine monophosphate (GMP) and purine xanthine. Upregulation of this purine salvage aids cell proliferation [23].

## 3. Discussion

In UM, liquid biopsies can detect the presence of the primary tumor and metastases by measuring certain biomarkers [24]. With adequate biomarker levels, even driver mutations or copy number variation profiles can be derived. However, primary UMs are small, and the amount of tumor-derived material in the blood is low at diagnosis [6]. As a result, these liquid biopsy techniques can be inadequate for the detection and prognostication of UM. Thus, other modalities should be explored, and one of these biomarkers is metabolites. Metabolites represent a biological phenotype, and minimally invasive metabolomics shows the potential to detect dysregulated processes. Differences in metabolite abundancies in peripheral blood are already used as biomarkers and can reveal patients harboring a malignancy and detect the upregulation of alternative energy consumption or inflammatory responses associated with cancer [11,13,25,26]. In addition, recent studies describe the potential of plasma-derived metabolomics for the early detection of malignancies [11,27]. In this pilot study, we report a difference in the metabolome of UM patients and controls, and annotated metabolites are indicative of associations with malignant processes. Metabolites represent the phenotype of the body, and changes in metabolite abundance can occur due to diet, lifestyle, and the time of day [28]. Therefore, our promising results in this pilot study warrant further investigations in the plasma metabolome of UM patients in a larger cohort with, ideally, the patients’ partners as controls to eliminate these possible biases. The observed difference in metabolite profiles between UM patients and controls could allow for confirmation of small UMs and should thus be a focus of future research.

Participants in our control cohort were patients of our outpatient clinic who received care for ocular diseases commonly found in the elderly that also occurred in our cohort of UM patients. The controls were, on average, older than UM patients; however, we did not find systemic differences in metabolic abundance in the controls of older age, nor did we find outliers based on age. Plasma was extracted from the blood of patients and controls within hours after blood withdrawal, and plasma was subsequently stored at −80 °C. The average storage times of plasma derived from controls and UM patients were similar, but we did observe a difference in the range of storage times. These storage time differences could affect metabolite concentrations in individual samples as the concentration of certain metabolites can be affected during storage [29,30]. However, other studies showed that long sample storage does not systemically affect the used biomarkers [31,32]. The possible differences in metabolite concentrations due to storage time differences should be reduced as the median of these storage times is similar. Additionally, unsupervised two and three-dimensionality reduction analyses did not reveal outlier clusters based on harbored diseases or storage time. Furthermore, we normalized to correct for technical variation during the untargeted metabolomics analysis. Nonetheless, technical differences could possibly arise from other sources, and in future experiments, the use of more similar, unaffected controls may confirm our results. For this pilot study, we were interested in differences between metabolite abundance patterns in large, prognostically relevant UM patient groups Therefore, we selected our patients with typical characteristics regarding age, tumor location, primary and secondary driver mutations, CNV profiles, and metastatis-free survival. In our cohort, samples with sufficient quality that were stored and processed in a similar way with information addressing all the before-mentioned characteristics were relatively rare. Thus, several interesting questions were not addressed and could be answered in follow-up studies with larger cohorts. One of these questions is evaluating the metabolome of patients who contracted a UM on earlier age compared to older age, as age is an important factor for metabolite patterns [33]. Ideally, this would identify similar metabolite abundance patterns that distinguish UM patients from controls in the older population, broadening the reach for an early-stage diagnosis of UM. We have used a discovery and replication cohort to validate the found metabolite abundance differences in UM and controls. This means that even though we included 110 peripheral blood samples of UM patients, the subgroups within our cohorts are relatively small. Metastatic risk in UM is associated with mutations in *BAP1, SF3B1*, and *EIF1AX*, and one of these three genes is typically mutated in UMs. Next to the typical mutations, UMs with rare driver mutations that are associated with metastatic risk are reported. For example, deleterious *MBD4* mutations are associated with a higher tumor mutation burden and a hypermutator phenotype [34]. UMs with *MBD4* deficiency are interesting, as these tumors respond to immune checkpoint inhibitors [35]. Using a large dataset of 1099 UM patients, the calculated prevalence of germline *MBD4* mutations in UM cases was 0.7% [36]. Furthermore, somatic *MBD4* mutations are uncovered in a small subset of metastatic UM [37]. And although this gene is very interesting, we could not obtain sufficient samples to analyze MBD4 as a subgroup. Next to the secondary driver mutations, differential expression of the genes *FOXD1* [38], *HES6* [39], *ABHD6* [40], and *PRC1* [41] is associated with metastasis formation in either *BAP1* or *SF3B1*-mutated UMs. Unfortunately, we do not have transcriptomic information on most of the selected patients in this study. Another subgroup we did not address comprises patients with unknown secondary driver mutations, as for this pilot study we first wanted to elucidate metabolite abundance patterns for the typical prognostic UM subgroups. To counteract the small sample numbers and unknown additional prognostically relevant mutations, we have evaluated metabolite abundance patterns in patients that would eventually develop metastases and patients that would not develop metastases. We could not distinguish prognostic subgroups using untargeted metabolomics from peripheral blood withdrawn at the moment of diagnosis using the original batches and ten bootstrapped, in silico, repeated analyses of randomly picked samples, of which 50% were evenly distributed within the cohorts. And although our clinical follow-up, including survival, was known for at least 60 months, the samples were taken at the time of diagnosis, and metabolic changes discriminating molecular subclasses and prognostically relevant subgroups might not have reached a detectable level at the early stage of disease.

The metabolic phenotype changes due to cancer [42] and several affected pathways can be detected in peripheral plasma. Only a subset of metabolites is currently annotated and corresponding to proteins, hampering the translation from metabolites to proteins and biological pathways. Therefore, it is striking to see that the indicated affected pathways are associated with cancer. In UM, our metabolomics data suggest the upregulation of tRNA charging, which promotes proliferation, metastasis, and invasion of malignant cells [43]. Furthermore, consistent with oncogenic processes, decreased adenine and guanine suggest either decreased purine degradation or increased salvage of purine bases. Several metabolite studies in UM focused on metastatic cell lines [44,45,46]. BAP1-mutated cell lines show the upregulation of oxidative phosphorylation, pyruvate dehydrogenase, and succinate dehydrogenase subunit A (SDHA) [44,45]. In lung cancer, the TCA cycle is restored by the upregulation of SDHA and valine metabolism [47]. Dysregulation of SDHA, oxidative phosphorylation, and pyruvate dehydrogenase was not observed in patients’ plasma as only a subset of metabolites was annotated, but valine was less abundant in UM patients compared to controls, consistent with an upregulation of the valine metabolism. Additionally, the upregulation of Arginase 1, which catalyzes the hydrolysis of arginine into ornithine and urea, in UM patients was reported by Velez et al. [25]. Our data also suggest a lower abundance of arginine and citrulline in the plasma of UM patients, in line with an upregulation of Arginase 1 [25].

## 4. Materials and Methods

### 4.1. Experimental Design

This retrospective observational study evaluated the discriminatory potential of metabolite patterns in uveal melanoma patients compared to control participants. Furthermore, we aimed to distinguish mutation subgroups of uveal melanoma patients by metabolic patterns obtained from peripheral blood withdrawn at the time of diagnosis.

### 4.2. Patient Selection

Patients were selected based on their secondary driver mutation (*EIF1AX* (n = 24), *SF3B1* (n = 33), or *BAP1* (n = 56)) and availability of peripheral blood prior to treatment, and they were age and sex matched. Survival data and tumor characteristics were recorded; additionally, liver function tests and medical imaging were performed routinely at intervals of six months for the screening of (hepatic) metastases. Furthermore, known confounding diseases (such as cardiovascular disease, diabetes, or pulmonary disease) were collected and noted from patient information next to medication use.

The patients enrolled in the Rotterdam Ocular Melanoma Study Group (ROMS) between 1998 and 2021 (medical ethics committee (MEC)-2009-375) (Table 1). Treatment for the primary tumor and phlebotomy was performed at Erasmus MC (Rotterdam, the Netherlands) or the Rotterdam Eye Hospital (Rotterdam, the Netherlands). Age- and sex-matched non-UM controls (n = 46) consisted of patients receiving care for ophthalmic diseases (Appendix A) in Erasmus MC who had peripheral blood withdrawn prior to treatment between 2016 and 2017 (Biobank Combined Ophthalmic Research Rotterdam Biobank (CORRBI) (MEC-2012-031)) (Appendix A).

### 4.3. Collection of Blood

Blood was collected in 8 mL lithium-heparin blood collection tubes between 1998 and 2021. Within 6 h after blood collection, plasma was separated from red and white blood cells by centrifuging once at 3500× *g* at 4 °C for 10 min. The supernatant was centrifuged a second time at 17,000× *g* at 4 °C for 10 min and subsequently, the supernatant was collected and stored at −80 °C. The average storage time per sample did not differ between the individual groups of mutations (range 0–20 years) and controls (range 3–5 years), but the range varied.

### 4.4. Liquid Chromatography-Mass Spectrometry

UHPLC Orbitrap MS was performed on plasma from peripheral blood, as described in R. Bonte et al. [20]. UHPLC-MS analyses were performed using a Dionex Ultimate 3000 UHPLC chromatograph (Thermo Fisher Scientific, Waltham, MA, USA) and a Q Exactive Plus hybrid quadrupole-orbitrap mass spectrometer with a heated electrospray source (Thermo Fisher Scientific, Waltham, MA, USA). The injection volume was 3 µL for all samples. A flow rate of 400 µL/min was used for chromatographic elution. UHPLC-MS sample analyses were performed in positive and negative ion modes, as some metabolites are only detected in one of the two modes. Capillary voltage in the negative ion mode was set to −3.5 kV and 3.5 kV in the positive ion mode, and the capillary temperature was set at 380 °C, and the auxiliary temperature was set at 300 °C during analyses. UHPLC sample analyses in both positive and negative ion modes were preceded by four injections of water and a quality control sample; hereafter, the samples were run in a randomized order.

To eliminate systemic variation, internal and external standard mixtures were added, and feature abundance was normalized based on the abundance of these internal and external standards. The internal mixture contains: D5—L-Phenylalanine (600 µmol/L), [13C]—Thymidine (300 µmol/L), 1,3-15N—Uracil (300 µmol/L), D10—Isoleucine (500 µmol/L), D6—Ornitihine (225 µmol/L), D4—Tyrosine (230 µmol/L), 5-bromo-DL-tryptophan (85 µmol/L), 3,3-dimethylglutaric acid (300 µmol/L), D4—glycochenodeoxycholic acid (44 µmol/L), D3—Carnitine (285 µmol/L). The external mixture contains D3—methylmalonic acid (100 µmol/L), D2—Uridine (207 µmol/L), D8—L-valine (670 µmol/L), D2—acetylcarnitine (45 µmol/L), D3—hexanoylcarnitine (19 µmol/L), D3—tetradecanoylcarnitine (19 µmol/L), D3—hexadecanoylcarnitine (6 µmol/L).

### 4.5. Metabolomics Analyses

All feature abundancies were log-transformed to log(1 + x) and Z-transformed (after normalization), after which they followed a Gaussian distribution. All non-zero values were pooled (across all features and samples). From this pool, the 2nd percentile was determined. A feature was included when at least 5 samples had an abundance above the 2nd percentile. Furthermore, for the merged dataset, only features that were detected in both batches were considered.

#### Removing Inter—And Intra-Batch Variation

Two different normalization methods were used to correct for inter- and intra-batch variations. For the analysis of the replication and discovery set, all features were normalized using a linear regression (BayesianRidge method from Scikit-learn [48]) on the first two principal components of the internal standards. This results in abundancies that are corrected by the overall abundancies of the internal standards. However, this normalization step was only performed on features where 75% of the samples had an abundance above zero. Furthermore, prior to fitting the model, outlier samples were removed when sample |Z-score| > 3. Note that features for which less than 75% of the samples had an abundance above zero were not normalized, and thus their original (log-transformed) abundance was used in the analysis. For the merged dataset, we used Metchalizer (v1, https://github.com/mbongaerts/Metchalizer), accessed on 3 March 2023, with an initial log transformation (log(x + 1)). Four metrics were used to judge the performance of normalization: WTR-score (within variance total variance ratio), quality control correlations, batch prediction score, and QC prediction score, as described by Bongaerts et al. [14].

### 4.6. Statistical Analysis

To test for differences in storage time, the Kruskall–Wallis test was performed as the data was not normally distributed. The difference in age of blood withdrawal was tested by the student *t*-test, as the data was normally distributed. Tumor characteristics were tested using a one-way ANOVA. These tests were performed using GraphPad Prism 9 for macOS, version 9.4.1, San Diego, CA, USA.

For unsupervised clustering, the PCA and t-SNE dimensionality reduction methods from Scikit-learn were used to explore the potential clustering of the four groups. Afterwards, for supervised clustering, the PLS-DA from Scikit-learn was used to assess the clustering of the four groups based on m/z feature abundance. An RFC from Scikit-learn [48] containing 150 decision trees with a maximum depth of 100 was trained for classifying UM patients versus non-UM controls and for classifying the prognostically relevant groups based on metabolite patterns. We used a LOOCV strategy to assess the performance of the classifier. For each validation round, one sample from the dataset was left out and put into the test set (i.e., the leave-one-out procedure) (Appendix A). Since not each group is equally represented in the dataset, classification biases might occur for groups with more samples. To correct for this potential bias, we oversampled each group such that each group had an equal number of samples (n_i_ = 200). Normal noise N(0, 0.25) was added to the dataset to prevent overfitting, which serves as regularization. For each sample, we obtained a ‘probability score’ of the correct class using the ‘predict_proba()’ method from the RFC. ROC curves for UM patients versus control participants were created using these ‘probability scores’. In order to estimate the variance in the area under the curve (AUC), a bootstrap with replacement was used. In total, 25 bootstraps were performed, from which the average and standard deviation of the AUC were determined. Additionally, we repeated this leave-one-out cross-validation (LOOCV) procedure ten times, using 50% of the random samples for each (sub)group, followed by the same steps as described above (Appendix A). In that way, for each statistic, we were able to obtain 10 values, which indicates the robustness of our findings.

Ingenuity Pathway Analysis (version 01-20-04) was used for analyzing differentially regulated pathways using annotated metabolites [49].

## 5. Conclusions

We showed that UM patients can be distinguished from controls in a near-perfect manner using plasma metabolite patterns at the time of diagnosis, which could shorten the patient delay. Typically, UM-like lesions are detected by optometrists, family doctors, or ophthalmologists. These healthcare providers then refer patients to specialized tertiary centers, where UMs are diagnosed. A large prospective cohort study in the UK comprising 2384 UM patients showed that the time of referral can vary and that a delay in diagnosis is common. They also showed that 23% of the UM cases were initially missed, resulting in more enucleations [7]. This small pilot study highlights the potential of untargeted metabolomics to support the diagnosis of UM and could prioritize the referral of patients based on a UM risk profile to minimize patients’ morbidity. This warrants a follow-up study that includes larger sample sets and preferably uses the partners of UM patients, from whom blood will be withdrawn simultaneously.

## Figures and Tables

**Figure 1 ijms-24-05077-f001:**
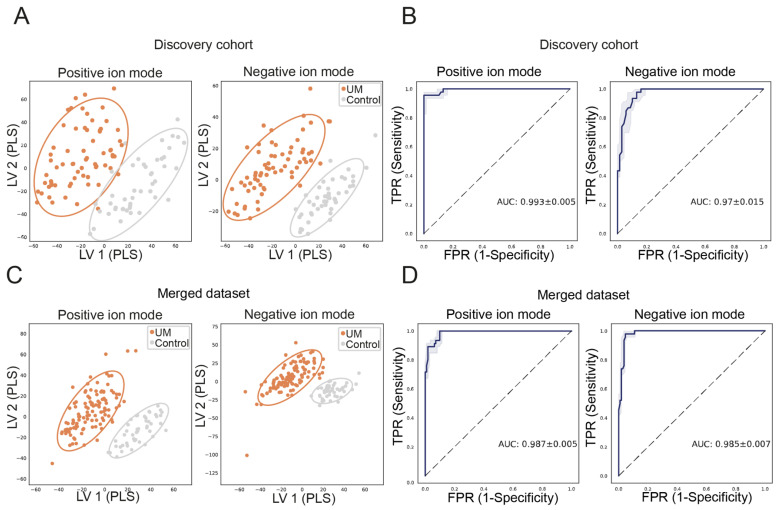
Distinct metabolite patterns in UM patients can be used for differentiation. Mass/charge features (with distinct retention times) were determined using ultra-high performance liquid chromatography mass spectrometry, and these features were used for supervised dimensionality reduction methods and to train a random forest classifier (RFC). The partial least squares determinant analysis shows separation of UM samples compared to samples of controls in the positive and negative ion modes, in the discovery cohort (**A**) and merged dataset (**C**), in which UM patients are depicted in orange and controls are shown in gray and the circles represent the 95% confidence interval. We have trained an RFC on the metabolite patterns derived from UM patients and controls. The blue fitted line is the receiver operating characteristic (ROC) curve that represents the performance of our RFC model, as evaluated using the leave-one-out cross-validation procedure, by sensitivity on the *Y*-axis and 1-specificity on the *X*-axis for the discovery cohort (**B**) and merged dataset, respectively (**D**) with a 25–75% interval depicted in grey. The overall test performance is shown by area under the curve (AUC), and an AUC greater than 0.7 is generally considered a threshold for fair performance whereas an AUC of 1 indicates a perfect performance. The AUC of our test in the discovery cohort is 0.99 and 0.97 in the positive and negative ion modes, respectively, which means a near-perfect ability to differentiate metabolite patterns of UM patients compared to controls. The AUC of 0.99 and 0.99 for the positive and negative ion modes in the merged dataset indicate a robust differentiating power of our test when samples are added.

**Figure 2 ijms-24-05077-f002:**
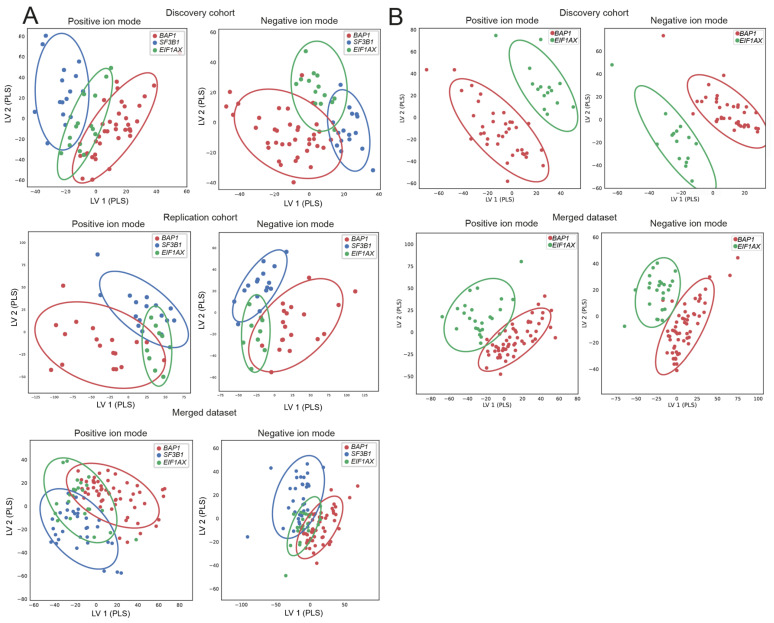
Supervised dimensionality reduction analyses show separation in molecular subclasses. The partial least squares determinant analysis (PLS-DA) shows moderate separation of the molecular subclasses based on the metabolite patterns in the discovery cohort, replication cohort, and merged dataset (**A**). Additionally, the prognostically poor and favorable secondary driver mutations, *BAP1* and *EIF1AX*, show good separation based on the supervised dimensionality reduction analysis (**B**). However, training a random forest classifier to distinguish the different subgroups using a leave-one-out procedure led to poor classification performances (Table 3).

**Table 1 ijms-24-05077-t001:** Cohort description. Cohort characteristics of discovery and replication cohorts used in this study. Significant differences between discovery and replication cohorts are depicted as followed: * *p* < 0.05; *** *p* < 0.001, **** *p* < 0.0001. UM patients are divided into molecular subclasses and are age and sex matched. The primary driver mutation, T-class, tumor size, inflammatory infiltrate, and ciliary body involvement were comparable. Necrosis, closed vascular loops, and epithelioid tumors were more prevalent in *BAP1*-mutated tumors. Metastatis-free survival (MFS) is in months; longest tumor diameter (LTD) and prominence are measured in millimeters. Abbreviations used: SD: standard deviation; CI: confidence interval; NE: not evaluated; Ns: non-significant difference; MFS: metastatis-free survival; LTD: longest tumor diameter; D: discovery set; R: replication set; y: years; mo: months; mm: millimeters; CB: ciliary body.

		*BAP1* n = 56		*SF3B1* n = 33		*EIF1AX* n = 24	
		D (n = 37)	R (n = 19)	D (n = 16)	R (n = 17)	D (n = 15)	R (n = 9)
Age at onset (y) Ns	Mean (SD)	68.8 (10.7)	64.4 (12.9)	54.3 (12.2)	58.6 (14.7)	61.3 (12.0)	62.0 (12.1)
Gender	Male	17	12	7	8	8	5
	Female	20	7	9	9	7	4
MFS (mo) ***	Mean (SD)	30.0 (29.2)	38.1 (36.6)	61.7 (46.5)	73.0 (61.1)	70.6 (61.2)	78.6 (37.5)
Primary driver mutation	*GNAQ*	7	10	5	7	5	4
	*GNA11*	10	9	5	10	5	5
	*CYSLTR2*	2	0	0	0	0	0
	Missing	18	0	6	0	5	0
Tumor location	Choroid	32	16	13	17	14	9
	CB	5	3	3	0	1	0
T-class	1	3	1 66	2	2	2	1
	2	11	6	6	8	5	3
	3	18	10	7	6	7	5
	4	5	1	1	1	1	0
	Missing	0	1	0	0	0	0
LTD (mm) Ns	mean (SD)	13.7 (3>3)	13.9 (3.0)	14.2 (3.5)	13.3 (3.0)	12.5 (3.4)	12.0 (3.6)
Prominence (mm) Ns	mean (SD)	8.6 (4.1)	7.8 (3.5)	6.9 (3.6)	6.3 (2.3)	7.1 (2.3)	7.9 (3.1)
Cell type ****	Epithelioid	9	2	1	0	0	1
	Spindle	3	1	7	12	9	6
	Mixed	21	16	7	5	6	1
	NE	4	0	1	0	0	0
Inflammatory infiltrate	Yes	3	7	2	4	0	1
	No	4	12	3	13	4	8
	NE	30	0	11	0	11	0
Necrosis *	Yes	13	9	4	5	4	1
	No	20	10	10	12	9	8
	NE	4	0	2	0	2	0
Closed vascular loops *	Yes	20	15	5	3	4	1
	No	12	4	10	14	9	6
	NE	5	0	1	0	2	2
CB involvement	Yes	13	7	6	2	3	0
	No	19	12	8	15	10	7
	NE	5	0	2	0	2	2

**Table 2 ijms-24-05077-t002:** Validation of classifiers. Leave-one-out cross-validation of trained random forest classifiers that classify UM samples versus control samples in the discovery cohort (**A**) and merged dataset (**B**), as these datasets contained control samples. Accuracies of 0.88 and 0.94 are observed in the discovery cohort in the negative and positive ion modes, respectively. In the merged dataset, the accuracies were 0.91 in the negative and 0.93 in the positive ion mode.

(A)	Discovery Cohort UM Patients versus Controls						
Negative Ion Mode				Positive Ion Mode			
	Precision	Recall	F1-Score		Precision	Recall	F1-Score
UM	1.000	0.794	0.885	UM	0.955	0.941	0.948
Control	0.767	1.000	0.868	Control	0.915	0.935	0.925
Accuracy	0.877			Accuracy	0.939		
**(B)**	**Merged datasets UM patients versus controls**						
	**Negative ion mode**				**Positive ion mode**		
	**precision**	**recall**	**F1-score**		**precision**	**recall**	**F1-score**
UM	0.990	0.875	0.929	UM	0.955	0.946	0.951
Control	0.763	0.978	0.857	Control	0.872	0.891	0.882
Accuracy	0.905			Accuracy	0.930		

**Table 3 ijms-24-05077-t003:** No differential metabolic patterns in prognostic subgroups. A random forest classifier (RFC) was deployed to investigate differential metabolite patterns in UM subclasses in the discovery cohort (**A**), replication cohort (**B**), and merged dataset (**C**); and the prognostically relevant subgroups, *GNA11* versus *GNAQ* (**D**), *BAP1* versus *EIF1AX* (**E**) and, ultimately, the formation of metastases (**F**). Leave-one-out cross-validation shows poor discriminatory power of the random forest classifiers based on metabolite abundancies in these analyses.

A	Discovery Cohort UM-Subtypes and Controls						
	Negative Ion Mode				Positive Ion Mode		
	Precision	Recall	F1-Score		Precision	Recall	F1-Score
*BAP1*	0.532	0.676	0.595	*BAP1*	0.453	0.649	0.533
*SF3B1*	0	0	0	*SF3B1*	0	0	0
*EIF1AX*	0	0	0	*EIF1AX*	0	0	0
Control	0.714	0.978	0.826	Control	0.8	0.957	0.871
Accuracy	0.614			Accuracy	0.6		
**B**	**Replication cohort UM-subtypes**						
	**Negative ion mode**				**Positive ion mode**		
	**precision**	**recall**	**F1-score**		**precision**	**recall**	**F1-score**
*BAP1*	0.652	0.79	0.714	*BAP1*	0.7	0.778	0.737
*SF3B1*	0.556	0.625	0.588	*SF3B1*	0.474	0.563	0.514
*EIF1AX*	0	0	0	*EIF1AX*	0.75	0.333	0.462
Accuracy	0.568			Accuracy	0.605		
**C**	**Merged datasets UM-subtypes and controls**						
	**Negative ion mode**				**Positive ion mode**		
	**precision**	**recall**	**F1-score**		**precision**	**recall**	**F1-score**
*BAP1*	0.565	0.64	0.6	*BAP1*	0.581	0.705	0.637
*SF3B1*	0.367	0.289	0.324	*SF3B1*	0.5	0.289	0.367
*EIF1AX*	0.41	0.321	0.36	*EIF1AX*	0.444	0.286	0.348
Control	0.78	1	0.876	Control	0.763	0.978	0.857
Accuracy	0.601			Accuracy	0.618		
**D**	**Merged datasets primary driver mutation**						
	**Negative ion mode**				**Positive ion mode**		
	**precision**	**recall**	**F1-score**		**precision**	**recall**	**F1-score**
*GNA11*	0.571	0.571	0.571	*GNA11*	0.571	0.571	0.571
*GNAQ*	0.609	0.609	0.609	*GNAQ*	0.609	0.609	0.609
Accuracy	0.591			Accuracy	0.591		
**E**	**Merged dataset *BAP1* and *EIF1AX* mutations**						
	**Negative ion mode**				**Positive ion mode**		
	**precision**	**recall**	**F1-score**		**precision**	**recall**	**F1-score**
*BAP1*	0.571	0.632	0.6	*BAP1*	0.636	0.778	0.7
*EIF1AX*	0.417	0.357	0.385	*EIF1AX*	0	0	0
Accuracy	0.515			Accuracy	0.538		
**F**	**Merged dataset metastatic formation of UM patients**						
	**Negative ion mode**				**Positive ion mode**		
	**precision**	**recall**	**F1-score**		**precision**	**recall**	**F1-score**
Yes	0.5	0.15	0.231	Yes	0.333	0.1	0.154
No	0.726	0.938	0.818	No	0.71	0.917	0.8
Accuracy	0.706			Accuracy	0.676		

## Data Availability

Our ethics committee does not allow sharing of individual patient or control genotype information in the public domain. Data and code can be provided by E.B., pending scientific review. Requests for the data and/or code should be submitted to the corresponding author (E.B.).

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
