# Peer review of "Uveal Melanoma Patients Have a Distinct Metabolic Phenotype in Peripheral Blood"

_ijms, 2023, doi:10.3390/ijms24065077_

Round 1
Reviewer 1 Report
Comment 1: This is a very interesting study to show Uveal melanoma patients have a distinct metabolic phenotype in peripheral blood. The experiments are well designed and the statical analysis of the results are nicely shown and described. The observations may help for early-stage diagnosis of uveal melanoma patients.
Comment 2: In this study why author selected only samples from 37 BAP1, 16 SF3B1, 15 EIF1AX patients? Why do authors not consider another gene for uveal melanoma patients? many studies reported other genes involved in uveal melanoma.
Comment 3: Have the author compare younger age (20-45-year-old UM cases, very rare) vs older age (46-80-year-old) UM cases. Observed any differences between these groups?
Minor suggestions
Comment 4: The author should include the latest references. For example, Paul Hofman, Prognostic Biomarkers in Uveal Melanoma (2022), and others recently published articles.
Comment 5: Recheck and correct the figure number in the result section
Author Response
Dear reviewer,
Thank you for your kind comment regarding the potential of using the metabolic phenotype of uveal melanoma patients for early-stage diagnosis. We want to express our gratitude for your valuable comments and thorough review of our manuscript. We included 113 peripheral blood samples of uveal melanoma patients with BAP1, SF3B1, or EIF1AX mutations. Other, less prevalent, genes that are associated to metastatic risk were not included in this study as we would risk losing power to deploy a machine learning algorithm looking at metabolite abundancy differences in a replication cohort and validation cohort. Nonetheless, you have a compelling argument that uveal melanomas comprise more than only the BAP1, SF3B1, and EIF1AX mutations. We have added a section addressing other known mutations and genes with differential expression that are associated with metastatic risk in the discussion. Specifically, deleterious MBD4 mutations are reported to have prognostic influence. Sadly, we do not have enough patients in our cohort for statistical power, as uveal melanomas with this mutation are scarce. Therefore, we did not include patients harboring MBD4 mutations in this study. If we would detect more MBD4 mutations in uveal melanoma patients, we will take the prognostic subgroup into account in follow-up studies. In addition to the mutation groups, your comment on the subgroup of young uveal melanoma patients is an interesting question. Unfortunately, sample numbers in our cohort are also insufficient to analyze differentially abundant metabolite patterns in this subgroup. We have addressed this in the discussion section. Thank you for the suggestion to cite more recent published articles. We have included your suggestion as well as other recent published articles. Our apologies for the mistake in result figure numbers and thank you for noticing, we have addressed this mistake and corrected it.
Thank you again for your extensive review of our manuscript.
Reviewer 2 Report
The authors apply an untargeted metabolic approach to determine metabolic patterns in the peripheral blood of uveal melanoma patients with different secondary driver mutations compared to control patients.
The paper is interesting since metabolic changes in the peripheral blood are gaining increasing attention as potential biomarker in several cancer types including cutanous melanoma.
However, some improvements should be made to deem this paper suitable for publication:
Line 89: The authors should change "slightly older" since the two cohorts differ significantly in age.
The mentioned sample numbers of the replication cohort in section 2, line 83 do not match with the numbers indicated in Table 1.
In section 2.3, the authors mention that they observed a low abundancy of the amino acids arginine, asparagine, cysteine, leucine, lysine, methionine, phenylalanine, proline, tryptophan, tyrosine, and valine, a higher abundancy of glycine, as well as a lower abundance of adenine, guanine, and D-ribose-5-phosphate in UM patients without showing the data. I would like to see at least a list with annotated metabolites and respective p-values or a heat map.
Suggesting that arginase 1 is upregulated based on a lower abundance of arginine and citrulline is highly speculative. Since not only the superpathway of citrulline but also other pathways seem to be differentially affected in UM patients, I suggest not to focus on arginase 1 in the title of section 2.3..
Numbering of supplementary Figures S4-S8 does not match with cross-references in the manuscript.
Author Response
Dear reviewer,
We want to express our gratitude to you acknowledging the potential of metabolic patterns in blood of uveal melanoma patients. We thank you for your thorough review.
Thank you for pointing out our textual mistakes. We have revised ‘slightly older’ and removed “slightly”, as controls are, indeed, significantly older than the patients. We have reviewed the sample number errors and corrected this in section 2, line 83. The numbering of supplementary figures is corrected, thank you for noticing our mistake.
Thank you for suggesting including a list of differentially abundant annotated metabolites and we have included a list with annotated metabolites, human metabolite database IDs, p-values obtained from the t-test between UM and controls and fold-change (supplementary excel file). These differentially abundant metabolites were used for the ingenuity pathway analysis and we have referred to it in the main text and figure S12 accordingly. Additionally, we have included the differentially abundant unannotated features from the t-test of tumor versus controls in the supplementary excel file.
We should not have emphasized on arginase 1, as this is speculative. Therefore, we have changed the title of section 2.3.
Thank you again for your extensive review of our manuscript.